# Introduction of a New Test Methodology for Determining the Delayed Cracking Susceptibility

Anton Hopf [1,*], Moritz Klug [1], Kürşat Durmaz [1], Klaus Goth [1] and Sven Jüttner [2]

1   Mercedes-Benz Group AG, 70372 Stuttgart, Germany
2   Faculty of Mechanical Engineering, Institute for Materials and Joining Technology, Otto-von-Guericke Universität Magdeburg, Universitätsplatz 2, 39106 Magdeburg, Germany
*   Correspondence: anton_sampiero.hopf@mercedes-benz.com

**Abstract:** A missing test methodology that allows for the determination of delayed cracking susceptibility of laser welds of high-strength sheet steel is presented. Unlike other cold crack testing methods, this test is based on a self-restraint testing of specimens welded from thin sheet materials without welding consumables and external loading. The potential test procedure with sample geometry, clamping device and documentation of the cracks is described. It is shown that the position of the weld on the specimen is a critical parameter and the susceptibility to cold cracking increases with increasing edge distance. The test methodology in combination with the most critical seam position is successfully used to rank two different steels regarding their susceptibility to delayed cracking. Further investigations are conducted evaluating the cold cracking susceptibility at different energy levels and lubricating conditions. It is proven that the lubrication has a significant influence on the susceptibility to cold cracking. Nevertheless, a narrow but safe process window is found.

**Keywords:** AHSS; laser welding; delayed crack; HACC; self-restraint test; body-in-white production




## 1. Introduction

In order to achieve environmental, political, and economic goals, modern car manufacturers have been facing the challenge of using lightweight components on the entire body-in-white structure, since the beginning of the 1990s. To address this issue, Advanced High Strength Steels (AHSS) with a yield strength of over 800 MPa have been developed by steel makers, allowing for the reduction of the material thickness in automotive parts [1]. In order to combine high formability requirements in addition to high strength and crash safety, complex manufacturing processes, which eliminate the use of cost-intensive alloy concepts, have also been introduced in recent years [2]. Laser beam welding (LBW) is widely used among thermal joining processes in automotive production due to its advantages in terms of efficiency, quality, and flexibility [3]. The higher process speeds, the need for only one-sided accessibility of the components to be joined as well as the numerous seam geometries, and the concentrated heat input strengthen the use of laser welding compared to conventional thermal joining processes, such as resistance spot welding (RSW) and gas metal arc welding (GMAW) [4]. The welding of AHSS in combination with the different alloying components and the brittle microstructure after laser welding due to high cooling rates leads to the fact that these materials are considered more susceptible to delayed cracking during thermal joining than conventional steels.

Cold cracks occur preferentially in welding areas with low ductility and high hardness. In general, this type of cracking occurs as a result of a combination of brittle microstructures, high tensile stress, and increased proportions of diffusible hydrogen [5], which is illustrated in Figure 1.

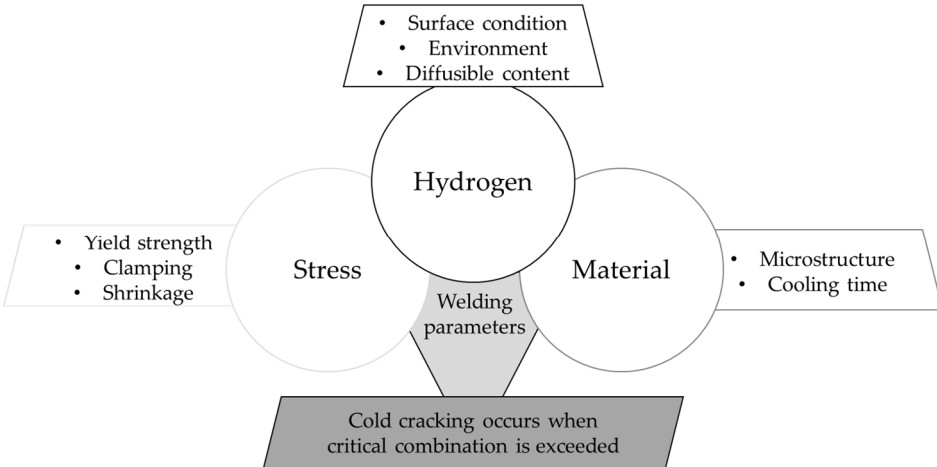

**Figure 1.** Main factors influencing cold cracking according to EN ISO 17462 [6].

The mechanism of delayed cracking formation in arc [7–9] and laser welds [10–12] has already been investigated in earlier studies. It has been shown that the diffusible hydrogen content in laser and arc welds increases with the additional use of corrosion protection or forming oil [11]. In automotive production, due to the rare use of welding consumables, delayed cracking has rarely been detected. However, large amounts of hydrogen can diffuse into the weld bead through welding without shielding gas [13]. In general, cracks in the weld seam are considered critical, which often results in the specification of a crack-free welded joint [14]. With the present uncertainty on how safe and reliable welding processing of innovative steel materials take place, a useful testing methodology for ranking the cold cracking susceptibility of steels is of acute importance. In the following, the review literature on the various cold cracking test methods for welds is additionally extended by recent investigations. The missing test methodology for testing laser-welded overlap joints is revealed and a new test method that allows the investigation of delayed cracking phenomenon in laser welding is presented.

## 2. Cold Cracking Test

The development of a test methodology that allows the ranking of welded steel joints with regard to the susceptibility to cold cracking is of decisive importance. In addition, there is a lack of detailed research on the cold cracking phenomenon itself, i.e., a testing method for laser welding of thin sheet steels.

### 2.1. Common Cold Cracking Tests

For determining the tendency to cold crack formation of weld seams, a large number of test methods have already been introduced. These are weldability tests that evaluate the susceptibility of base materials and welding consumables to cold cracking with regard to the material and welding process. The cold cracking test methods can be divided into qualitative tests, i.e., evaluating whether or not crack occurs and quantitative tests (e.g., limit value for crack-free welds). Kurji et al. give detailed instructions for developing test methods for welded specimens with regard to susceptibility to cold cracking [15].

In addition to subdividing the tests according to their result, the test methods can also be distinguished in terms of the type of load applied. Self-restraint cold crack tests are based on welded specimen without external influence of an applied force. The level of load from self-restraint tests depends on internal stresses and strains generated by the welding process as well as the geometrically determined specimen stiffness [16]. While this type of tests is simple and inexpensive to perform, it typically involves complex and non-uniform loads that develop during cooling which cannot be easily quantified. This limits the information gained from the tests, which would help defining critical conditions required for cracking. The externally loaded cold cracking tests, on the other hand, are carried out by applying an

external mechanical load, which superimposes the weld-specific residual welding stresses. External load levels can be imposed independently of the welding parameters. The load can be chosen to reflect the practical application as realistically as possible, i.e., in terms of the yield strength of the material, residual stress and shrinkage.

The most common cold cracking test methods are standardized in EN ISO 17642 Parts 1–3 [6,17,18]. In addition, Kannengiesser et al. give an overview of the large number of cold cracking tests [16]. Nevertheless, these tests are applied to a sheet thickness greater than 6 mm, which exceeds the standard sheet thicknesses used in car body construction, which is up to 4.5 mm. Thus, these test methods cannot be applied properly for automotive welding processes [19]. Furthermore, the standards mainly apply to arc welding with significantly larger weld bead dimensions than those found in common laser welding processes.

A first attempt to transfer cold cracking testing by implant test to laser welding application was conducted by Sievers et al., whereby the difficulties caused by the original test concept for large-volume welding baths predominated. Thus, the transfer was not possible without further modifications [20]. Consequently, there is still the need to establish a suitable test method for investigating the susceptibility to cold cracking of laser welded thin sheet overlap joints in automotive body-in-white constructions.

The most commonly used cold cracking tests from EN ISO 17642 are summarized in Table 1. Kannengiesser et al. and Kurji et al. have also reviewed and evaluated further test methods [15,16].

**Table 1.** Overview of standardized cold cracking tests from EN ISO 17642 [6].

| Test | Load Type | Joining Method | Weld Shape/Type | Material Thickness | Aim of the Test | Transfer to Automotive Applications |
|---|---|---|---|---|---|---|
| Controlled Thermal Severity-Test | Self-restraint | GMAW | Fillet weld | >6 mm | Cold cracking susceptibility | Restricted |
| Tekken | Self-restraint | GMAW | Y joint | >6 mm | Determination of the lowest welding energy, the lowest preheating temperature, the lowest holding temperature, the highest diffusible hydrogen content. | Restricted |
| Lehigh Test | Self-restraint | GMAW | U joint | >6 mm | | No |
| Implant | External | GMAW | Bead on plate weld | >20 mm | Determination of the lowest welding energy, the lowest preheating, interlayer and holding temperature, the highest diffusible hydrogen content, the critical stress | Restricted |

### 2.2. Recent Studies on Cold Cracking Tests

Numerous studies have already shown the embrittlement potential of welded joints made of high-strength thin sheets. These studies focused on the subsequent loading of the welded joint with applied force in a corrosive environment [21,22]. This does not satisfy the requirement of a weldability test with regard to cold cracking tendency.

Loidl has investigated the suitability and potential applicability for high-strength steel sheets in the body-in-white production steps from delivery, forming and joining to cathodic dip painting and corrosion, with different test methods [23]. The examined self-restraint Cup-in-Cup Test promisingly combines forming and resistance spot welding processes. Due to difficulties caused by variations from sample tolerances, which makes test reproducibility and result interpretation almost impossible, the test turns out to be unsuitable. The investigation of spot-welded plates with externally applied load in the static shear tensile test was able to determine differences in the materials at different service lives, but this testing method instead investigates the cold cracking susceptibility of welded material under the influence of corrosive media and not the risk of cracking due to the

welding process itself. The so-called wedge test was intentionally designed to investigate the influence of different wedge geometries, representing different stress states on the weld while spot welding. Due to the unfavorable specimen geometry, this test is also not expedient, since the gap condition in the vicinity of the weld point leads to corrosion failure depending on the medium.

Testing the cold crack sensitivity of arc welded joints by means of a four-point bending test has already been extensively and successfully demonstrated [8,10]. Schwedler et al. have shown that non-destructive testing by acoustic emission technology can be advantageous in obtaining quantitative results, such as the time to crack appearance [8]. The possibility of adjustable load application allows a quantitative comparison of the welded specimens but requires the necessary testing and bending equipment. Maeda et al. demonstrated that X-ray inspection is another potential detection method for cold cracks [12]. Recent attempts on developing a cold cracking test method for LBW and thin sheet steels are summarized in Table 2.

**Table 2.** Overview of recent studies on cold cracking tests.

| Test | Load Type | Joining Method | Weld Shape/Type | Material Thickness | Input Factors | Transfer to Automotive Applications |
|------|-----------|----------------|-----------------|--------------------|---------------|-------------------------------------|
| 4 Point Bending Test | External | GMAW, RSW, LBW | Overlapping, fillet weld | <2 mm | Preforming, hydrogen content, heat input, force | Restricted |
| Welded Shear Load Test | External | GMAW, RSW, LBW | Overlapping, fillet weld | <2 mm | Preforming, hydrogen content, heat input, force, loading speed | Restricted |
| Cup-in-Cup Test | Self-restraint | RSW, LBW | Overlapping | <2 mm | Preforming, hydrogen content, heat input | Restricted |

In conclusion, a test methodology for repeatable and comparable testing of the susceptibility to cold cracking in laser-welded sheets has not been established yet. Therefore, a practicable self-restraint cold crack testing methodology for assessing the risk of hydrogen-assisted delayed cold cracking during laser welding of AHSS and Ultra High Strength Steels (UHSS) in the thin sheet area is presented, giving a detailed process description and presenting numerous variable parameter variations.

## 3. Test Method

The cold cracking test method, which has been developed, allows the characterization of the cold cracking sensitivity of high strength sheet materials, such as AHSS and UHSS. Based on its simple sample geometry and test procedure, this test method allows for distinguishing both the cold cracking susceptibility of different alloy grades from one supplier and the cold cracking susceptibility of the same alloy grade from different suppliers very fast and accurately. It is therefore suitable for material approval tests with regard to weldability and enables materials to be ranked in terms of their susceptibility to cold cracking.

### 3.1. Sample Geometry and Test Procedure

The sample geometry with a length of 105 mm and a width of 45 mm is used to characterize the cold cracking susceptibly is shown in Figure 2.

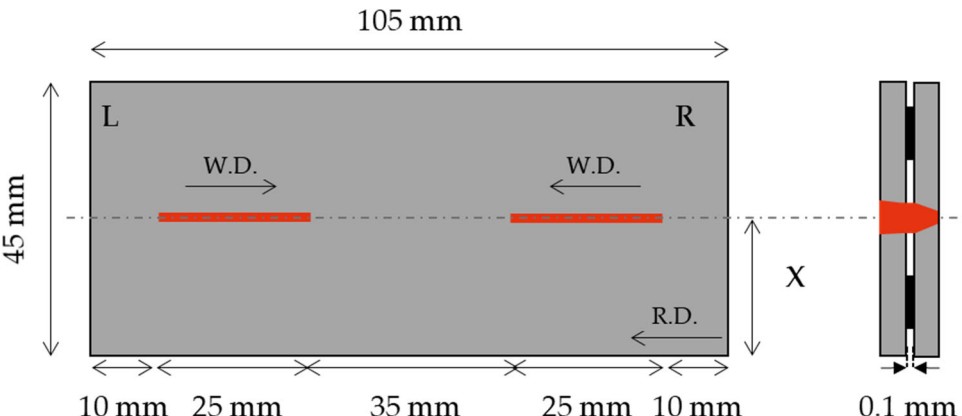

**Figure 2.** Fully overlapping sample geometry (W.D. for welding direction and R.D. for rolling direction).

This sample geometry was developed in accordance with SEP1220-3 (Testing and Documentation Guideline for the Joinability of thin sheet of steel—Part 3: Laser beam welding) and can be produced using common sheet metal cutting processes, such as shear cutting, water jet cutting, laser cutting, wire erosion, etc. [24]. Differing from the test procedure according to SEP1220-3, two pieces of the test samples are fully overlapped, and then the laser welding of two seams are carried out in such a way that the end crater section of the weldments remains in the center of the sample. As this test procedure demonstrates a self-restraint test, the clamping conditions need to be kept identical during welding. Nevertheless, the clamping device according to the SEP1220-3 test procedure is highly recommended, since it enables a fast, stable, and standardized clamping condition during welding [24].

As coated steels are primarily the focus of automotive applications, the degassing of the coating layer has to be ensured during laser welding to avoid weld imperfections. This can be assured either using spacer foils as described in SEP1220-3 or a special dimple geometry, which was introduced by Reiniger et al. [24–26].

There are different ways for crack detection and inspection, such as visual inspection using a magnifying glass or stereomicroscopy. Most of the cracks are already visible with 10 times magnification, which makes this inspection method practical and useful. Figure 3 demonstrates an example of a cracked weld seam seen from the top and bottom surface.

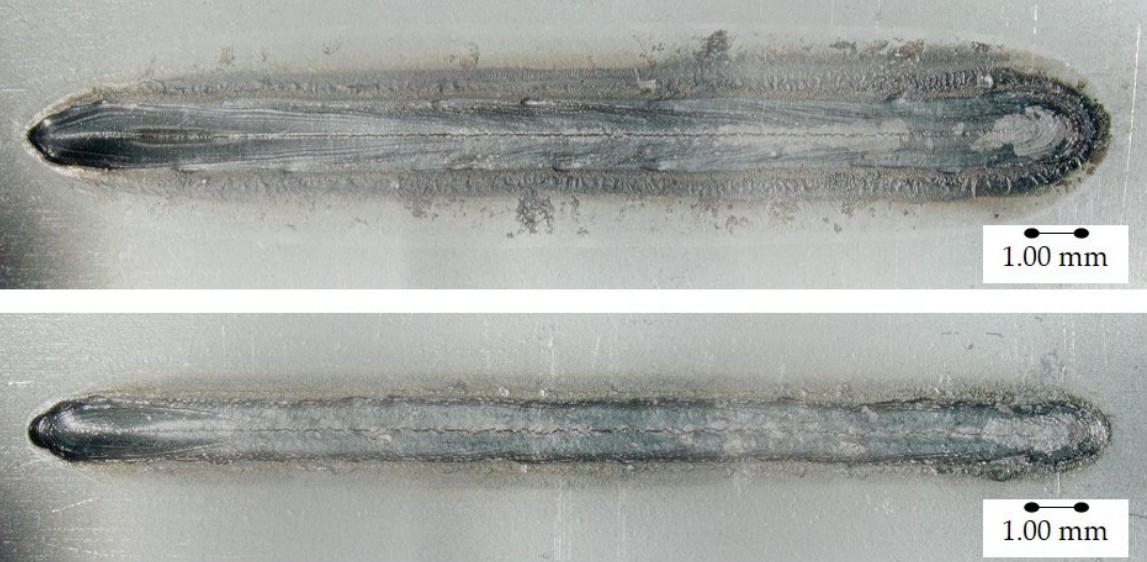

**Figure 3.** View of the weld crack from upside (**top**) and downside (**bottom**).

In addition, further investigations either destructive or non-destructive, such as metallographic investigations or X-ray testing, can be conducted to gain more information. Existence of the cracks or the crack length as well as the crack growth rate can be determined as test criteria. However, the existence of the cracks is a useful criterion which helps to determine a statistical cracking probability, since it can be detected very fast without using extensive evaluation methods.

### 3.2. Testing Parameters

The critical factors affecting the cold cracking phenomenon are well known and shown in the prior section. The proposed cold cracking test method allows the systematic evaluation of these factors and helps understanding the effect of these on cold cracking susceptibility.

### 3.2.1. Material

Due to its simple test geometry, any alloy grade in sheet form can be investigated using this test method, regardless of whether the material is coated or not. It allows the use of different specimen and stack thicknesses or combinations for both similar and dissimilar joints. In addition, testing of greater material thicknesses with the same geometry in the form of weld beads on plates is also possible. Thus, this specimen geometry and test method can be utilized to investigate the susceptibility to cold cracking of high strength structural materials that are prone to delayed cracking, such as fine-grained structural steels.

### 3.2.2. Hydrogen

This test procedure offers various ways to investigate the effect of hydrogen on delayed cracking formation of laser welds. First, the material can be welded after a proper surface cleaning to evaluate the effect of the total hydrogen amount of the material, both trapped and diffusible hydrogen, on delayed crack formation. This enables very rapid evaluation and elimination of materials which, depending on the manufacturing process in the steel mill, have a high hydrogen content in their microstructure. The other way is the precharging of samples with defined diffusible hydrogen, i.e., [21]. Third, the effect of the external hydrogen sources can be simulated using carbon hydride-based lubricants or oils, which are commonly used in sheet metal forming processes [11,12]. For evaluating the material specific critical hydrogen value that is introduced by lubricants, the test can be carried out using the same material in the same welding conditions but different amounts of lubricants.

### 3.2.3. Critical Stress Condition

Different studies showed the formation of stresses in weldments during and after the welding process [27,28]. The main reasons for the formation of the welding stresses are heat input and high cooling rate after welding, which cause the thermal expansion and contraction in welds that vary depending on the material properties, such as thermophysical properties, and clamping conditions.

The introduced test method allows for varying the heat input by means of welding speed, laser power, spot diameter, etc., during laser welding of the materials, and helps therefore to understand the tendency of the cold crack formation with respect to the weld bead shape and penetration. Furthermore, it enables the determination of the material-specific critical edge distance at which the material tends towards the cold crack formation. The latter has an impact on the cooling conditions after welding as well as restraint conditions [29].

### 3.3. Schematic Testing Procedure

This test method enables of testing a variety of factors and understanding their influence on the susceptibility of a particular material to the formation of cold cracks. It provides an opportunity to investigate which modifications need to be made for defects avoidance in weld design. A summarized example of a test procedure is shown in Figure 4.

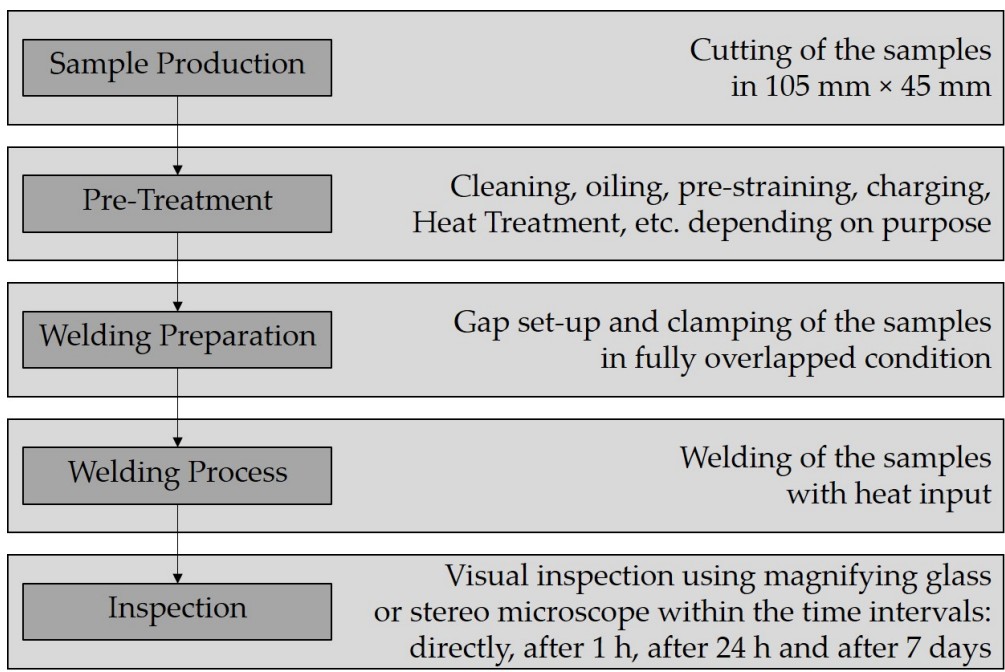

**Figure 4.** Exemplary steps of the cold cracking test procedure.

In the following section the optimal weld seam position for dedicated evaluation of the cold cracking susceptibility of steels is studied for the presented test method. With this knowledge the main goal of a delayed susceptibility ranking between different steel grades is demonstrated. Part of the study also includes investigations on the cracking behavior regarding different surface conditions and welds with increasing penetration depths.

## 4. Materials and Experimental Procedure

Two commercially available hot dip galvanized AHSS according to VDA239-100 [30], CR700Y980T-DH (steel A) and CR700Y980T-DP (steel B), were used in the experiments. Mechanical properties of the materials are listed in Table 3. The corresponding chemical compositions and the carbon equivalents according to Yurioka are shown in Table 4 [31]. The carbon equivalents are calculated with the following equations:

$$CEN = C + A(C) [Si/24 + Mn/6 + Cu/15 + Ni/60 + (Cr + Mo + Nb + V)/5 + 5 \cdot B] \quad (1)$$

$$A(C) = [0.75 + 0.25 \cdot \tanh(20 \cdot (C - 0.12))] \quad (2)$$

**Table 3.** Mechanical properties of steels A and B.

| Steel | $R_{p0,2}$ (MPa) | $R_m$ (MPa) | $A_g$ (%) | $A_{80}$ (%) | $r_{2-20}$ | $n_{2-20}$ |
|---|---|---|---|---|---|---|
| A | 838 | 1034 | 8.16 | 12.5 | 0.812 | 0.102 |
| B | 712 | 1040 | 7.2 | 10.7 | 0.599 | 0.104 |

**Table 4.** Chemical composition (wt%) and carbon equivalent of steels A and B.

| Steel | C | Si | Mn | P | S | Al | Cr | Mo | Ti | Nb | B | Cu | Ni | V | Fe | CEN |
|---|---|---|---|---|---|---|---|---|---|---|---|---|---|---|---|---|
| A | 0.15 | 0.94 | 2.47 | 0.006 | 0.000 | 0.220 | 0.11 | 0.01 | 0.005 | 0.001 | 0.000 | 0.037 | 0.040 | 0.002 | bal. | 0.573 |
| B | 0.08 | 0.47 | 2.27 | 0.011 | 0.003 | 0.044 | 0.32 | 0.20 | 0.028 | 0.033 | 0.001 | 0.017 | 0.034 | 0.002 | bal. | 0.382 |

All specimens from steels A and B were of 1.5 mm thickness and were shear cut to the size of 45 mm × 105 mm. After cutting, the specimen surfaces were cleaned with acetone. To ensure degassing of the zinc coating while welding the upper-side surface of

the bottom sample was dimpled, as described in Section 3.1. Half of the samples were additionally coated with 3 g/m² of forming lubricant, formulated with mineral oil and anti-rust additives (Oest, Platinol B 804/3 COW-1).

A Yb:YAG disc laser with a maximum power of 8 kW (Trumpf, TruDisk 8001) and a programmable focusing optic (Trumpf, 3D PFO) was used both for dimpling and welding of the samples. The focal spot was 0.6 mm in diameter. The laser beam was used adjusted laterally. Compressed air pressed through the crossjet nozzle protected the safety glass of the optics from fume and spatters during welding, and no shielding gas was used. For clamping, the proposed device from SEP 1220-3 was used with pneumatically controlled copper clamps with a free center width of 20 mm and a slotted bottom plate of 15 mm width. The overall set-up with all process steps from dimpling (a and b) over clamping (c) to welding (d) is shown in Figure 5.

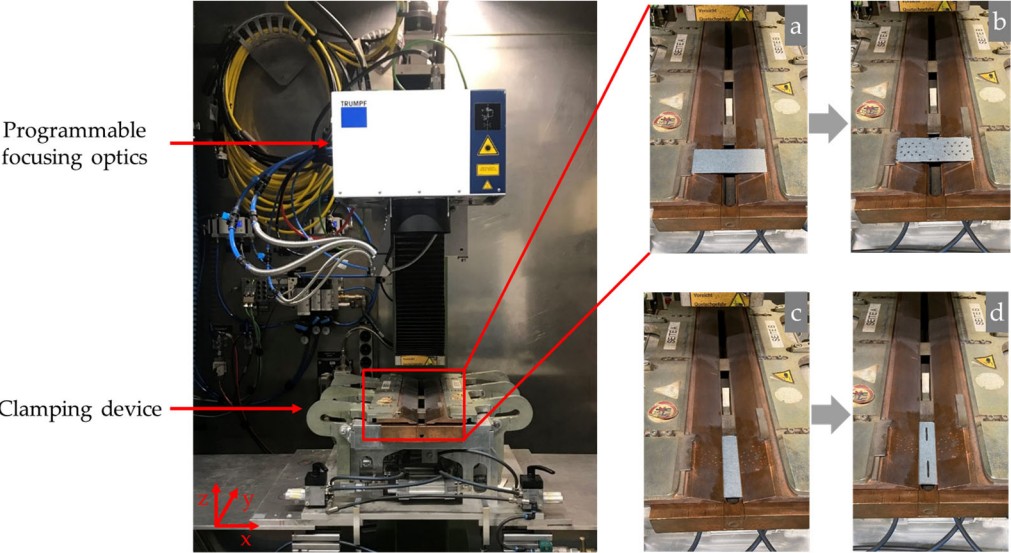

**Figure 5.** Welding and clamping set-up and dimpling and welding steps.

The fully overlapping joints were welded according to Figure 2 with a constant laser power of 5.6 kW. For determination of the most suitable weld seam position as well as the comparison of different steel grades regarding their cold cracking sensitivity, all tests were carried out at a constant speed of 65.3 mm/s, which was determined according to SEP1220-3 and ensures a full penetration of the lap joint. The standardized procedure in SEP1220-3 specifies the welding depth regardless of material and sheet thickness. The weld lengths were 25 mm per weld. In total, 10 samples each with two welds were welded for every parameter variation, creating 20 welds with the same parameters.

The corresponding line energies are calculated with the following equation:

$$E = P/v \tag{3}$$

with P for laser power and v for welding speed, respectively. The specimens were removed from the clamping device directly after welding. Visual inspection was carried out using a magnifying glass with 10× magnification according to Figure 4 directly after welding, after 1 h, after 1 day and after 7 days.

### 4.1. Determination of Critical Weld Seam Position

In order to investigate which seam position on the sample is most suitable for the differentiated analyzing of the susceptibility to cold cracking, different edge distances of the weld seam were investigated. All tests were carried out on steel A at a constant speed of 65.3 mm/s. A total of 10 specimens, each with two welds, were welded in lubricated condition with an edge distance of 5 mm, 10 mm, 15 mm and, in the center of the specimen,

at 22.5 mm, resulting in 20 welds for each edge distance. Figure 6 displays all different welded edge distances.

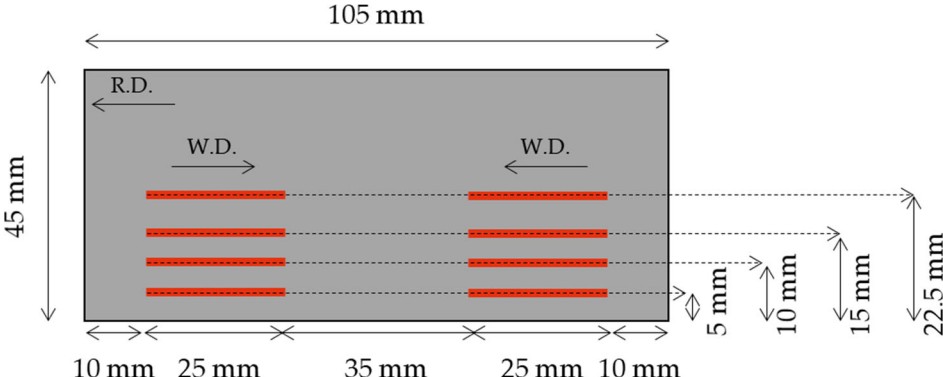

**Figure 6.** Edge distance positions for evaluation of the most susceptible position to cold cracking.

*4.2. Comparing Different Steels to Their Delayed Cracking Susceptibility*

Following the aim of the presented test method, both steel grades are investigated according to their susceptibility to cold cracking when welded with same parameters described above, but in lubricated condition. Therefore, the most critical weld seam position from Section 4.1. is used.

*4.3. Influence of Line Energy and Surface Condition on Delayed Cracking Susceptibility*

The fully overlapping joints were welded with the most critical weld seam position from Section 4.1 with a constant laser power of 5.6 kW and welding speeds ranging from 65.3 mm/s to 143.3 mm/s. In total, 10 samples each with two welds were welded in both cleaned and lubricated condition for each line energy, creating 20 welds with the same parameters. Table 5 displays all line energies used for welding.

**Table 5.** Overview of welding speeds and line energies used for welding depth studies with steel A.

| Name | Power (kW) | Speed (mm/s) | Line Energy (J/mm) |
|---|---|---|---|
| E5 | 5.6 | 143.3 | 39.1 |
| E4 | 5.6 | 113.3 | 49.4 |
| E3 | 5.6 | 90 | 62.2 |
| E2 | 5.6 | 83.3 | 67.2 |
| E1 | 5.6 | 75 | 74.7 |
| E0 | 5.6 | 65.3 | 85.7 |

## 5. Results

In this section, results from investigations on the effect of weld seam position, surface condition, and welding parameter on cold crack susceptibility are demonstrated. Moreover, the critical edge distance of weld seam which helps to classify different steels regarding to their cold cracking susceptibility is emphasized.

*5.1. Critical Weld Seam Position*

No cracks occurred at the edge distances of 5 mm and 10 mm from the longitudinal edge. A further increase in edge distance to 15 mm leads to a first delayed crack out of the 20 welds at this position. Moving the seam position to the middle of the sample raises the number of cracks to over 85%. Figure 7 shows the growing cracking occurrence with further distance to the sample edge at line energy $E_0$. In contrary, any investigated edge distance from 5 mm to 22.5 mm showed no cracks for steel B.

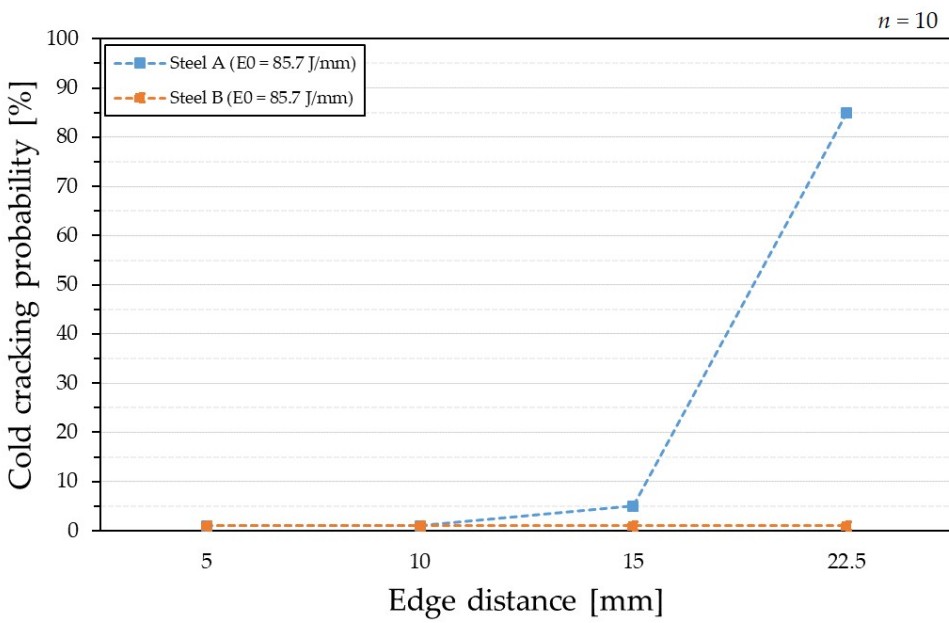

**Figure 7.** Critical edge distances of the weld seams to cold cracking for steels A and B.

## 5.2. Delayed Cracking Susceptibility of Different Steels

When comparing steels A and B, both welded with the same line energy at 22.5 mm edge distance, no cracks were detected for the cleaned samples. In contrast, the lubricated steel A showed 85% cold cracking sensitivity, while steel B still showed no cracks. The evaluation result when comparing both steels regarding their susceptibility to cold cracking is shown in Figure 8.

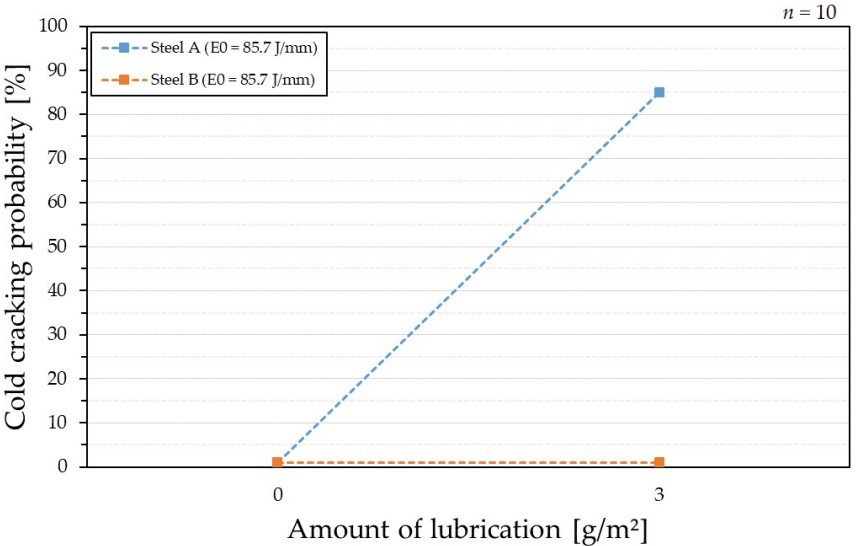

**Figure 8.** Comparison of the cold cracking probability between steels A and B in cleaned and lubricated condition.

Regardless of the different carbon equivalents of steels A and B, there is no difference in cold cracking probability in cleaned condition. Nevertheless, in lubricated condition cold crack formation occurs in steel A with the higher carbon equivalent compared to steel B.

## 5.3. Influence of Line Energy and Surface Condition on Delayed Cracking

All samples were welded at the most critical edge distance of 22.5 mm. Starting with the line energy that assures a minimum welding width of 1 mm on the top surface of the lower sheet, the stepwise reduction of speed leads to an increasing penetration of the two

steel sheets. The same results in terms of penetration depth were obtained regardless of the surface condition of the samples. The effect on the weld depth and material penetration is illustrated in Figure 9.

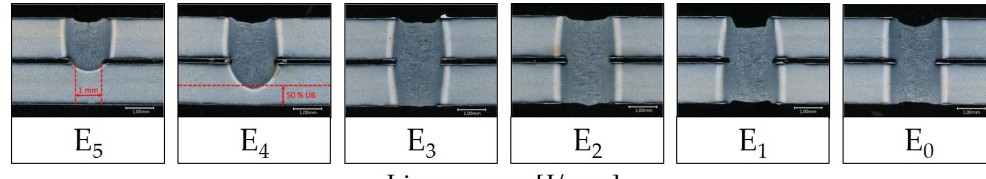

Line energy [J/mm]

**Figure 9.** Difference in depth penetration for steel A depending on line energy in cross-sections of lubricated samples.

In terms of cold crack probability, the results show differences between the two states regarding the surface condition of the samples. While the additional lubricating of the samples leads to cold crack formation depending on the line energy or penetration depth, cleaned samples did not show any cold crack formation. In total, 3 welds of the 20 least penetrated samples showed cracks. When the welding speed was reduced, the susceptibility to cracking also decreased to 0% at line energies of 62.2 J/mm and 67.2 J/mm.

From this point the trend is changing and further speed reduction leads to significant higher cold crack occurrence. A total of 7 of 20 welds showed cold cracks when the full penetration is achieved at the line energy of 74.7 J/mm. Additional energy input raises the probability to 85%, which is the highest value measured. Figure 10 displays the correlation between cold cracking probability linked to line energy used while welding for the two surface conditions.

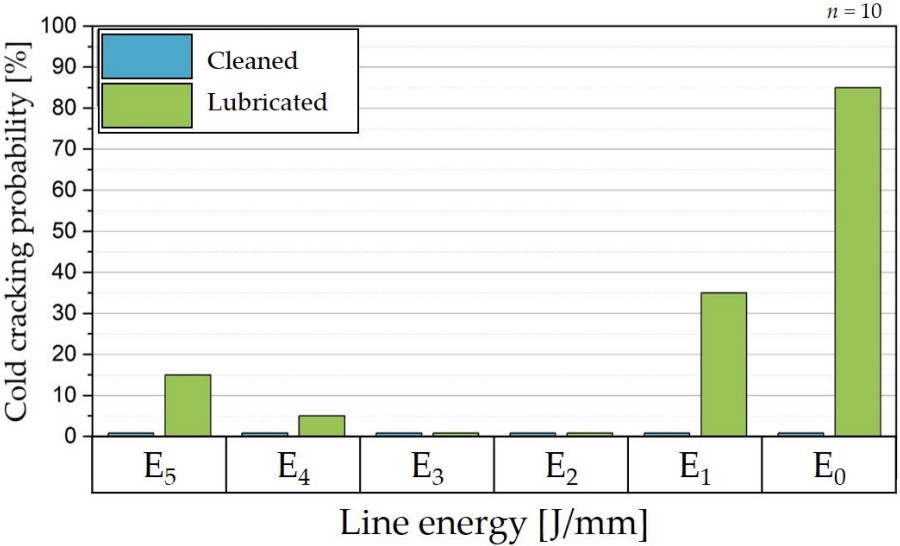

**Figure 10.** Difference in cold cracking probability of steel A depending on line energy and surface condition.

The samples were inspected at four different times with unchanging or raising numbers of cracks over time after the inspection directly after welding. Either no crack or no further cracks were evaluated at energies from 39.1 J/mm up to 67.2 J/mm. For the welding with the line energy of 74.7 J/mm directly after welding, no cold cracks were detected; thus, all cracking happened within the first hour. Most of the cracks detected at the line energy of 85.7 J/mm were already detected directly after the welding process. The crack probability increased by 10% within one hour up to 80%. After one day, 17 of the 20 welds were cracked and no other cracks were detected after one week. Figure 11 displays the time dependent cold crack formation depending on the heat input.

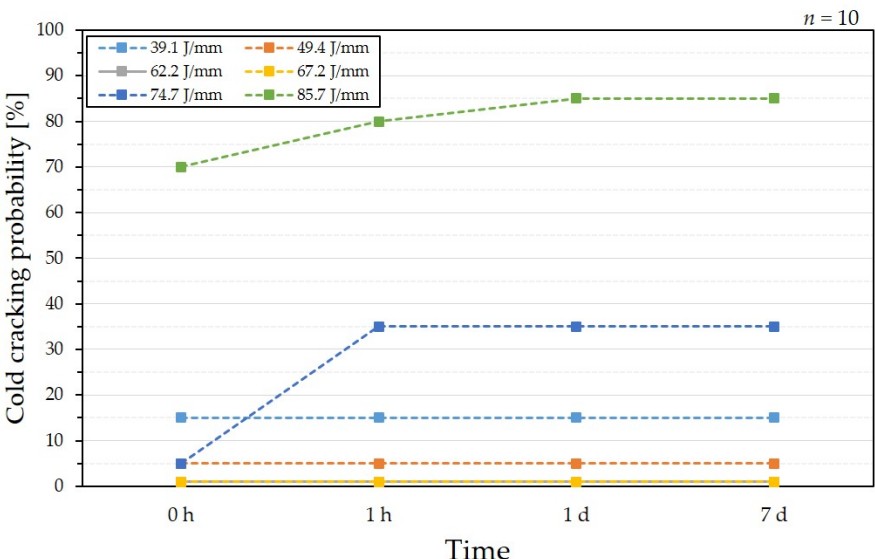

**Figure 11.** Cold crack formation of steel A depending on line energy and time.

## 6. Discussion

The cold crack susceptibility of AHSS in laser welding has not been investigated extensively yet. Maeda et al. and Jiang et al. reported that the high strength steels can exhibit cold cracking formation. However, there is a lack of a test method to evaluate this phenomenon in laser welding of sheet steels [12,32].

It is shown that the proposed cold cracking test method enables the ability to investigate the cold cracking susceptibility of thin sheets in laser welding. Furthermore, it is revealed that the cold crack formation of a particular material can vary depending on the heat input or penetration depth, and surface condition. While the cold crack formation can be avoided using the right parameter set, the safe process window is very narrow. The detected critical weld seam position shows high potential for determining the cold cracking susceptibility of any material and allows a ranking between different materials.

However, it has to be considered that the clamping conditions changed during the edge distance test due to the clamping setup. While close to the edge welding leads to one-sided clamping, an increase in edge distance allows for a two-sided clamping. Therefore, it has to be investigated whether the different cracking behavior results from the modifications in clamping conditions or from different cooling and shrinkage conditions due to the changed seam position.

As it is mentioned in the results, the carbon equivalent can be used as an indicator to distinguish materials' cold cracking susceptibility. However, differences in the surface conditions emphasize that the effect of hydrogen due to lubricant should not be neglected. It is shown that steel with a higher carbon equivalent is more prone to cold cracking when hydrogen is introduced through lubrication.

It also remains unclear which cause of cracking is mainly affected when the weld penetration depth is varied, since the change in heat input simultaneously changes the stresses and also the cooling conditions. On top, less lubricated surfaces are in direct contact with the keyhole, potentially causing less hydrogen uptake from the forming oil.

Beside many advantages of the presented self-restraint cold cracking test, there are some drawbacks of the test method. The amount of 20 equal welds per variation used in the presented study seems a reasonable amount for investigating the influence of single parameters. Thus, an open question remains about the statistical approach of the test method. For high informative value, a sufficient number of welds must be carried out to produce statically reliable results. The number of welds should be set as large as possible, but within reasonable limits.

With respect to the obtained crack growth rate, the intervals of inspection should clearly be discussed, because most cracks in this study were already present at the first evaluation point right after welding. Maeda et al. showed that most delayed cracks occurred within a few seconds to minutes after welding [12], representing a time of growth upstream of the presented inspection intervals.

Inspection with a magnifying glass also causes difficulties in determining the time-dependent crack length, due to missing constant monitoring options. Nevertheless, when the desired output is the cold cracking susceptibility of a material, the quantitative measure is sufficient.

## 7. Conclusions and Outlook

The necessity of developing a self-restraint cold crack testing method for laser welding of thin sheet AHSS from the current literature was demonstrated, and a new cold crack testing method with multiple parameter variation was successfully proposed.

In summary the following findings are the results of this study:

- The presented test method enables the ability to investigate the cold cracking susceptibility of laser welded thin sheet AHSS.
- It allows for ranking different AHSS with regard to their susceptibility to cold cracking after laser welding.
- Lubricating the sample surfaces clearly raises the risk of cold cracking after welding compared to cleaned samples.
- Cold crack formation is highly dependent on heat input, and could be avoided between 62.2 J/mm and 67.2 J/mm. Nevertheless, the safe process window is very narrow.
- Higher line energies tend to show more delayed cracking than cracking welds at lower energies.

With the presented cold cracking test method, further investigations on how different parameter settings cause cold cracking or how cracking could be avoided can be conducted. Besides the cold cracking test and possibility to compare different materials, future work has to be conducted to understand reasons behind the cold crack formation.

On the one hand, further metallurgical evaluations are crucial, such as microstructure analysis, hardness measurement, hydrogen measurements, fracture surface analysis, etc., for understanding the relations between material characteristics and welding parameters. On the other hand, it is also necessary to emphasize research on modern laser processing techniques, such as beam shaping and oscillation or (ultra-short) pulse beam to avoid the cold cracking of laser welds.

**Author Contributions:** Conceptualization, A.H. and K.D.; methodology, A.H., K.D. and M.K.; validation, A.H. and M.K.; formal analysis, A.H.; investigation, A.H. and K.D.; writing—original draft preparation, A.H. and K.D.; writing—review and editing, K.D., M.K. and A.H.; visualization, M.K. and A.H.; supervision, K.G. and S.J. All authors have read and agreed to the published version of the manuscript.

**Funding:** This research received no external funding.

**Institutional Review Board Statement:** Not applicable.

**Informed Consent Statement:** Not applicable.

**Data Availability Statement:** The data presented in this study are available on request from the corresponding author. The data are not publicly available due to restrictions regarding ongoing studies and agreements with third parties.

**Conflicts of Interest:** The authors declare no conflict of interest.

## Nomenclature

AHSS  Advanced High Strength Steel
E    Line Energy
GMAW  Gas Metal Arc Welding
P    Laser Power
LBW   Laser Beam Welding
RSW   Resistance Spot Welding
UHSS   Ultra High Strength Steel
v    Welding Speed
Yb:YAG  Ytterbium-YAG

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
