# Peer review of "Introduction of a New Test Methodology for Determining the Delayed Cracking Susceptibility"

_jmmp, doi:10.3390/jmmp7010026_

Round 1
Reviewer 1 Report
A novel test methodology that allows the determination of delayed cracking susceptibility of laser welds of high-strength sheet steel is presented in this study. This test method is based on a self-restraint testing of specimens welded from thin sheet materials without welding consumables and external loading. The results show that the weld position, the heat input and the lubrication have significant influence on the susceptibility to cold cracking. The novel test methodology for testing laser-welded overlap joints has a significant meaning about the evaluation of the delayed cracking in laser welding in the body-in-white production. However, a few points must be clarified before the paper can be accepted.
1. Please explain the reason why the cracking probability of steel A first decreased then increased with the line energy increased?
2. It is suggested to combine the result and discussion parts.
3.This study investigates the cracking tests of two kind of AHSSs, so the author should state what kind of material these conclusions are based on?
Author Response
- Thank you for addressing this issue. Further investigations have definitely to be done to understand this phenomenon and will be part of future work. In the scope of this paper, we want to present a new test method that allows different parameter variation. In the discussion part we mentioned the open topic: It remains also unclear which cause of cracking is mainly affected when the weld penetration depth is varied, since the change in heat input simultaneously changes the stresses and also the cooling conditions. On top less lubricated surfaces are in direct contact with the keyhole potentially causing less hydrogen uptake from the forming oil.
- With further remarks from other reviewers and some edits in the discussion part, we’d prefer leaving both chapters separated.
- The technical material names according to VDA239-100 have been added to the paper.
Reviewer 2 Report
It is important issue to address - the delayed weld crack formation - for new thinner materials in wide range of industries. different contributing factors are outlined and explained. the experimental data are presented as probability of crack formation dependent on treatment.
there are convincing differences for propensity of delayed crack formation dependent on surface treatment. it is obvious that oil was increasing probability of crack formation. in this regard discussion of surface hydrophylicity and hydropfobicity seems very relevant. water on surface before it can cause corrosion can contribute to hydrogen deposition to subsurface of metal (see the mechanism https://www.mdpi.com/2073-4344/11/9/1135). presence of water in gas or liquid phase can contribute to hydrogenation of metal.
another issue which might improve the qulity of weld is use of ultra-short laser pulses https://www.mdpi.com/2673-4591/11/1/44. heat deposition and stresses can be much better controlled using laser welding with short laser pulses. discussion of this issue can be very useful to add.
Author Response
- Addressing the first comment regarding the hydrogen deposition to subsurface, we want to mention the following paper (https://doi.org/10.1016/j.corsci.2011.05.023), which claims in the conclusion that the zinc layer acts as a diffusion barrier to the subsurface of the metal. All materials are stored at a dry location in-house at room temperature. Also all samples were cleaned before oiling and oiling took place immediately before welding.
- Thank you for addressing this issue. The aim of this paper is to demonstrate a new testing method which addresses cold crack formation in laser welds. For this investigation we worked with laser parameters/techniques according to SEP1220-3. The issue is mentioned in the added outlook. We are also working on crack avoidance strategies using different laser processing techniques. In future work we will definitely consider the mentioned approach using ultra-short-pulse lasers.
Reviewer 3 Report
A new test was developed for evaluating the delayed cracking susceptibility. The method and the index were introduced and discussed. The study is very interesting. However, some parts need to be modified.
1. The unit should be wrote in the figure, such as Fig.2. And the gap should be marked between upper and lower sheet in Fig.2. How to decide this small gap. In addition, for a high welding speed in Table 5, 25 mm of the length in weld seam seems to be short, so one weld seam could be obtained in less than 1 second. How to consider this affect?
2. In order to verify the accuracy and feasibility of this test method, some analyses should be done. Such as, the calculation of the carbon equivalent of the materials, and microstructure evaluation of the weld joint under different line energy should be shown and discussed to explain the change of the cold cracking probability.
Author Response
- The unit has been added in the figures. With respect to SEP1220-3 we decided to keep the gap equal to 0.1 mm from SEP. High welding speeds are an advantage of laser welding and this weld length is chosen according to the standard in SEP1220-3.
- The carbon equivalent according to Yurioka has been added to the paper. In the results as well as the discussion two paragraphs have been added to emphasize the materials differences. The aim of this paper is to demonstrate a new testing method which addresses cold crack formation in laser welds, further metallurgical evaluations (such as microstructure, hardness, hydrogen measurements, fracture surface analysis, etc.) have already been done, but are planned as part of another work/paper with focus on finding explanations of the cracking phenomenon. It is also mentioned in the outlook.
Reviewer 4 Report
A new test methodology that allows the determination of delayed cracking susceptibility is presented. The test methodology in combination with the most critical seam position is successfully used to rank two different steels. So this paper can be accepted and published under the major revision.
The comments are shown below:
1. Whether the new test methodology can be applied to other materials, such as aluminum alloy, titanium alloy.
2. What is the thickness of the sheet to use the methodology?
3. It is suggested to add schematic diagram when introducing cold crack test method.
4. This test method allows to distinguish the cold cracking susceptibility of different alloy grades. However, what are the disadvantages of this method?
5. When investigating the influence of line energy and surface condition on delayed cracking, what is selected of the edge distance.
Author Response
1. Q: Whether the new test methodology can be applied to other materials, such as aluminum alloy, titanium alloy.
A: The presented test method aims for investigations on AHSS used in automotive body-in-white structures. So far we are not familiar with occurance of cold cracks in laser welded aluminum parts. Due to it’s fcc crystal structure, we don’t expect to have this issue on aluminum alloys. Considering titanium further investigations could definitely be done, but the material is not commonly used in body-in-white production.
2. Q:What is the thickness of the sheet to use the methodology?
A: For our investigation we used steel sheets of 1.5 mm thickness, as mentioned in section 4. Also greater thicknesses or weld on bead samples can be tested with the methodology. Please see section 2.1: Nevertheless, these tests are applied to a sheet thickness greater 6 mm, which exceeds the standard sheet thicknesses used in car body construction which is up to 4.5 mm. Thus, these test methods cannot be applied properly for automotive welding processes [19].
3. Q: It is suggested to add schematic diagram when introducing cold crack test method.
A: For better understanding a section called 3 schematic testing procedure has been added. The aim of figure 4 was to show a schematic process diagram of our presented testing method.
4. Q: This test method allows to distinguish the cold cracking susceptibility of different alloy grades. However, what are the disadvantages of this method?
A: Actually we tried presenting some disadvantages in the end of the discussion section:
The amount of 20 equal welds per variation used in the presented study seems a reasonable amount for investigating the influence of single parameters. Thus an open question remains about the statistical approach of the test method. For high informative value, a sufficient number of welds must be carried out to produce statically reliable results. The number of welds should be set as large as possible, but within reasonable limits.
With respect to the obtained crack growth rate, the intervals of inspection should clearly be discussed, because most cracks in this study were already present at the first evaluation point right after welding. Maeda et al. showed that most delayed cracks occurred within a few seconds to minutes after welding [12], representing a time of growth upstream of the presented inspection intervals.
Inspection with a magnifying glass also causes difficulties in determining the time-dependent crack length, due to missing constant monitoring options. Nevertheless, when the desired output is the cold cracking susceptibility of a material, the quantitative measure is sufficient.
For clarification the following sentence has been added in the discussion section: Beside many advantages of the presented self-restraint cold cracking test, there are also some drawbacks of the test method.
5. Q: When investigating the influence of line energy and surface condition on delayed cracking, what is selected of the edge distance.
A: The selected edge distance is mentioned in section 4.2. and has been added to section 5.2 as well as 5.3.
Round 2
Reviewer 4 Report
It can be accepted in present form.